# Mental Health Support for Hospital Staff during the COVID-19 Pandemic: Characteristics of the Services and Feedback from the Providers

**DOI:** 10.3390/healthcare10071337

**Published:** 2022-07-18

**Authors:** Mélanie Loiseau, Fiona Ecarnot, Nicolas Meunier-Beillard, Alexandra Laurent, Alicia Fournier, Irene François-Purssell, Christine Binquet, Jean-Pierre Quenot

**Affiliations:** 1Service de Médecine Légale, CHU Dijon, Cellule d’Urgence Médico-Psychologique CUMP-21, 21000 Dijon, France; melanie.loiseau@chu-dijon.fr (M.L.); irene.francois@chu-dijon.fr (I.F.-P.); 2EA3920, University of Burgundy Franche-Comté, 25000 Besancon, France; 3Department of Cardiology, University Hospital Besancon, 25000 Besancon, France; 4Clinical Epidemiology/Clinical Trials Unit, Clinical Investigation Center, INSERM, CIC 1432, Dijon University Hospital, 21000 Dijon, France; nicolas.meunier-beillard@u-bourgogne.fr (N.M.-B.); christine.binquet@u-bourgogne.fr (C.B.); jean-pierre.quenot@chu-dijon.fr (J.-P.Q.); 5Laboratoire de Psychologie, Dynamiques Relationnelles Et Processus Identitaires (PsyDREPI), Université Bourgogne Franche-Comté, 21000 Dijon, France; alexandra.laurent@u-bourgogne.fr (A.L.); alicia.fournier@u-bourgogne.fr (A.F.); 6Service d’Anesthésie et de Réanimation, CHU Dijon-Bourgogne, 21000 Dijon, France; 7Service de Médecine Intensive-Réanimation, CHU Dijon-Bourgogne, 21000 Dijon, France; 8Équipe Lipness, Centre de Recherche INSERM UMR1231, 21000 Dijon, France; 9Espace de Réflexion Éthique Bourgogne Franche-Comté (EREBFC), 21000 Dijon, France

**Keywords:** mental health support services, COVID-19, health care workers

## Abstract

French authorities created mental health support services to accompany HCWs during the pandemic. We aimed to obtain feedback from staff providing these mental health support services within French hospitals to identify positive and negative features and avenues for improvement. A mixed-methods study was performed between 1 April and 30 June 2020. We contacted 77 centres to identify those providing mental health support services. We developed a questionnaire containing questions about the staff providing the service (quantitative part), with open questions to enable feedback from service providers (qualitative part). Of the 77 centres, 36 had mental health support services; 77.8% were created specifically for the epidemic. Services were staffed principally by psychologists, mainly used a telephone platform, and had a median opening time of 8 h/day. Thirty-seven professionals provided feedback, most aged 35–49 years. For 86.5%, it was their first time providing such support. Median self-reported comfort level was 8 (interquartiles 3–10), and 95% would do it again. Respondents reported (i) difficulties with work organisation, clinical situations, and lack of recognition and (ii) a desire for training. This study suggests that mental health support needs to be adapted to the needs of HCWs, both in terms of the content of the service and the timing of delivery.

## 1. Introduction

The pandemic caused by the SARS-CoV-2 virus has had devastating repercussions on populations worldwide. In France, this crisis has raised a number of thorny societal questions, calling into question the nation’s social and economic model. In the field of health, all groups of health professionals working in the hospital setting have been confronted with philosophical dilemmas about what is “good” or “right” and how best to optimise care for the greatest number while guaranteeing a dignified end-of-life, where necessary. In particular, regarding access to intensive care units (ICUs), health inequalities or inequities raised some veritable ethical dilemmas at the peak of the epidemic [1]. In view of the penury of ICU beds, the French Intensive Care Society issued guidelines to help physicians make decisions about how to manage severely ill patients and how to decide on admission (or not) to the ICU [1].

In a similar vein, but with a more general perspective, the National Consultative Ethics Committee proposed in March 2020 that Ethics Reflection Groups should be created within the already existing regional Ethics Reflection Centres (Espaces de Réflexion Éthique) [2]. In parallel to this proposal, and with a view to providing support to all healthcare professionals, the Committee also proposed the creation of mental health support services in hospitals. Indeed, it was already becoming clear in spring 2020 that healthcare workers (HCWs) from all professions were facing an unprecedented workload in a tense emergency context, with prevalent uncertainty in the workplace (i.e., fear of being contaminated) and at home (i.e., fear of contaminating one’s relatives) [3]. These mental health support services were intended to be modelled on existing Medico-Psychological Emergency Units, which are deployed in France after exceptional, large-scale emergencies such as terrorist attacks or natural disasters [4]. Usually, Medico-Psychological Emergency Units in France propose individual support and/or group debriefing in the immediate aftermath of traumatic events with a view to preventing chronic stress or post-traumatic stress disorder. However, these units were not designed to provide services outside the context of acute events, or on a scale commensurate with the mental health needs of a pandemic.

There was a compelling need for these new mental health support services, since several studies have since reported that there was significant mental suffering among HCWs, especially in the ICU setting [3,5]. This suffering was proportional to the epidemic intensity and tension within the hospitals, notably the number of available critical care beds [5,6,7]. Similar studies from previous epidemics, such as the Middle East Respiratory Syndrome virus outbreaks in the Middle East, have previously reported increased levels of mental stress and anxiety among HCWs faced with outbreaks of contagious disease [8].

The services were put in place to provide support and to accompany HCWs in their daily practice, with a view to proposing follow-up or specific interventions for those with nascent or persistent mental health disorders. To the best of our knowledge, no evaluations of this type of service have been published worldwide to date in the context of the COVID-19 pandemic, either regarding the modus operandi, the types of situations/cases dealt with, or the feedback from those providing this frontline mental health support. With a likely succession of epidemic waves in the relatively short term, we believe it is important to identify the advantages and disadvantages of this mental health support service to decide whether it would be useful to maintain it for the future.

Therefore, the aim of the present study was to obtain feedback from the staff providing these mental health support services within French hospitals during the COVID-19 epidemic to identify the positive and negative features of the initiative and to identify possible avenues for future improvement.

## 2. Methods

### 2.1. Study Design

The present mixed-methods study was performed between 1 April and 30 June 2020. It was a cross-sectional study comprising quantitative and qualitative enquiries. A graphical representation of the study methodology is provided in Appendix A.

#### 2.1.1. Identification of Centres Where Mental Health Support Services Were Provided

The PsyCOVID-ICU Study was conducted in 77 hospitals across France from 22 April to 13 May 2020. The methods and main results have previously been described elsewhere [5,7]. Briefly, the PsyCOVID-ICU study aimed to investigate the effects of the pandemic on the mental health of HCWs. The PsyCOVID-ICU study, including its quantitative and qualitative components and related sub-studies, received approval for all participating centres from the Ethics Committee of the French Intensive Care Society (N°20–33). The present study is a sub-study of PsyCOVID-ICU and focused on the staff providing mental health support to staff working in hospitals that participated in the PsyCOVID-ICU study during the first wave of the COVID-19 epidemic in France. In the main PsyCOVID-ICU study, an online questionnaire was distributed via the Limesurvey platform to 77 participating hospitals. In this questionnaire, one question asked about the availability of mental health support services for the staff in the hospital during the first wave of the epidemic. If respondents reported the presence of such services, there were further questions about when it was initiated, the type of services offered, the opening hours, how it functioned, etc. The English translations of the questions relating to mental health support services from the initial PsyCOVID-ICU questionnaire are provided in Appendix A.

#### 2.1.2. Contact with the Chiefs of the Identified Mental Health Support Services

In hospitals where mental health support services existed, we contacted the chief of the mental health support service by email to obtain detailed information about their service, its composition, its staff, and its daily functioning. When making contact by email to this end, we also provided the connection details to enable the participation of individual staff members. Reminders were sent to non-respondents one month after the initial contact.

#### 2.1.3. Contact with the Individual Staff Members Providing the Mental Health Support

Thirdly, we distributed an anonymous questionnaire online using the Limesurvey platform to the staff who provided the mental health support service. The questionnaire recorded information about the practical organisation and daily functioning of the service. The questionnaire was developed based on a review of the literature and a series of multidisciplinary meetings bringing together psychiatrists, clinical psychologists, sociologists, ICU physicians, and public health researchers. The questionnaire contained questions relating to the profile of the HCWs providing the service (sex, age, usual profession, prior experience (if any) of providing mental health support to HCWs). There were also a number of open questions to enable feedback from the service providers. The respondents were also asked to score their self-reported comfort level providing this service on a scale from 0 to 10, where 0 corresponded to “I was very uncomfortable providing this service” and 10 corresponded to “I was completely comfortable providing this service”. The English translation of the questionnaire is provided in Appendix A.

### 2.2. Participants

Inclusion criteria for centres were as follows: participation in the main PsyCOVID-ICU study; availability of mental health support services for HCWs in the centre. For centres meetings these criteria, the chiefs of the mental health support services were contacted. The chiefs who agreed to participate were provided with the connection details for the questionnaires, and the staff manning their mental health support services were also given the connection details to complete the online questionnaire about their perceptions and experience of providing this service.

### 2.3. Data and Analysis

Quantitative data are described as mean ± standard deviation (SD) or median (interquartile range), and categorical variables as number and percentage.

Free-text feedback was analysed by thematic analysis independently by two authors (M.L. (physician, MD) and F.E. (researcher, PhD); both female). Thematic analysis aims to describe major themes (which are spontaneously mentioned by all participants and developed at length) and minor themes (mentioned by some participants, and in less detail) by coding of discourse and grouping codes into meaningful categories [9]. Triangulation sessions were held with the full research team to reach consensus on the themes identified.

Data were managed using NVivo software (QSR International, Victoria, Australia). Quantitative data were analysed using SAS 9.4 (SAS Institute Inc., Cary, NC, USA). All interviews and free-text data were collected and analysed in French. All questionnaires were anonymous. Illustrative quotes were translated into English for the purposes of supporting the results. Respondents consented to the use of translated quotes from their questionnaires to illustrate scientific publications.

## 3. Results

### 3.1. Characteristics of the Mental Health Support Services

Of the 77 centres participating in the PsyCOVID-ICU study, a total of 36 had mental health support services in place, of which 28 (77.8%) were created specifically due to the COVID-19 epidemic. Eleven units (30.5%) worked in collaboration with the medico-psychological emergency unit of their hospital (Figure 1).

All the support services identified were staffed principally by psychologists (97.2%) and were mainly intended for use by HCWs (94.4%), although 75% of them reported that they were also available to help the families of COVID-19 patients. The majority (91.7%) of the support services identified used a central telephone platform and had a median opening time of 8 h per day (Table 1).

### 3.2. Contact with the Chiefs of the Existing Mental Health Services

Among the 36 centres that reported the existence of mental health services, all the chiefs of the mental health services were contacted. They explained the functioning of their service and their perception of the organisation of such services, especially the difficulty of creating mental health support services where none previously existed. They also reported that outside the context of the current crisis, they felt there was a need to design a mental health support service that could be deployed rapidly in times of crisis. Finally, they highlighted the under-use of the mental health support services made available for the COVID crisis, notably the telephone helpline. The chiefs reported that feedback from the potential users indicated dissatisfaction with the format (by telephone) and the service providers—namely that potential users within the hospital felt reluctant to call a mental health helpline when it was being manned by colleagues they might know from within the same hospital.

### 3.3. Feedback from the Service Providers

A total of 37 professionals staffing the support services provided feedback via the online questionnaires regarding their experience of providing mental health support during the epidemic (25 psychologists, 8 physicians, and 4 nurses; 28 females and 9 males). These professionals were mainly aged between 35 and 49 years old and usually worked in the greater Eastern region of France. They mostly had 10 years or more of experience. Only half had any training in the management of psychological emergencies, and 10.8% were specialists in occupational medicine (Table 2).

#### 3.3.1. Working in Mental Health Support Services

For the majority of respondents (86.5%), it was their first time providing mental health support services, especially by telephone. The median self-reported comfort level in providing this service was 8 [3,4,5,6,7,8,9,10], and 95% reported that they would be happy to do it again if needed.

Among the respondents, 15 HCWs reported that they felt very comfortable (score of 9 or 10 out of 10 on the self-reported comfort level) providing these services. These respondents were mainly psychologists (73%), and two-thirds of them had more than 10 years’ experience, while 8/15 (60%) were trained in emergency medico-psychological assistance or occupational health (1/15), thus predominantly comprising HCWs with the necessary professional competence.

#### 3.3.2. Difficulties Encountered

In response to the questionnaire, the participants recounted a total of 49 situations that they had encountered and found difficult during their time providing mental health support during the epidemic. We grouped these into two main themes, as follows: (i) difficulties with the work organisation and related to clinical situations and lack of recognition of their work; (ii) a desire for training and avenues for improvement.

The details of the number of times each theme was raised are given in Appendix A.

Difficulties with the work organisation

##### Organisation of the Service Delivery

The support services were, for the most part, created specifically for the purposes of the epidemic, where no such services had previously existed. The health authorities wanted to ensure a rapid response to the crisis. For the practical organisation, the mental health support services mobilised hospital staff who could no longer perform their usual duties due to redeployment of resources to deal with the epidemic. Thus, the support services were staffed by a disparate group of professionals from various backgrounds, without any clear hierarchical relationships. Most of them did not know each other and had never worked with the others before. This new and heteroclite collaborative group gave rise to a certain number of difficulties as they tried to establish a working relationship and clarify the hierarchical relations.

##### Unusual Nature of the Work

Due to the lockdown in force in France, the HCWs providing the mental health support services were mainly receiving calls over the telephone, which is a quite atypical manner of treating patients with mental health disorders. Some respondents found this difficult as they were missing a lot of non-verbal information that could be useful in a mental health consultation.

##### Uncertainty

The respondents unanimously mentioned the climate of uncertainty that reigned during the epidemic. Working during a healthcare crisis generated fear, not to say anxiety, because of the constant changes necessary to deal with the emerging disease and evolving knowledge. These adaptations were necessary at the level of the hospital but also among the HCWs providing the mental health support, with constant changes in the organisation.

Some HCWs were responsible for mobile teams (who go on callouts to respond on site) and found the burden of decision-making hard to bear. Indeed, they were responsible for deciding whether to engage a mobile team for the management of a patient, but this engagement could expose the whole team to a new and (at the time) relatively unknown disease.

Difficulties related to clinical situations

##### Refusal of Care and Under-Use of the Support Services

Many respondents to our questionnaire mentioned that they felt frustrated because the support services were not widely used (few, if any, calls). Alternatively, some felt frustrated because they had callers who refused to accept help. Examples cited included a resident with intensive psychological distress, who refused to accept help, or a physician with a panic attack who was too ashamed to ask for help. In view of this situation, some of the providers stated that they wondered about the relevance of the service, its modus operandi, and the type of help it offered. Several respondents felt that the service was not a good fit with the needs of the HCWs at the time.

##### Non-COVID-Related Problems

A number of respondents were surprised to find that they had callers whose problems were not COVID-related. Indeed, during their time working on the support services, several providers found themselves faced with clinical or socio-environmental problems that had been aggravated by the lockdown and were brought to a head, notably domestic violence or suicidal tendencies. Furthermore, some callers to the service had previous traumatic experiences or psychological disorders that were revived by the tense situation during the current epidemic, prompting them to call for help.

##### Lack of Recognition for their Work

The last difficulty mentioned was that the low uptake of the mental health support service may have mirrored a low level of recognition of the value of this service on the part of the hospital or on the part of colleagues. The HCWs providing the service reported feeling that their efforts were not adequately acknowledged.

Desire for training and avenues for improvement

Based on this new experience, which most providers nevertheless found enriching and satisfying, 58% of respondents declared that they would be glad to receive additional training with a view to repeating this activity. There was a general desire to improve personal skills in areas such as crisis management, managing stress, trauma, repeated bereavement, or the specificities of telephone interviews/consultations.

The question of the relevance of the service and how well it met the needs of its target audience was also underlined. The respondents were interested in attempting to improve the practical organisation and promoting collaboration between professionals from diverse backgrounds.

Several avenues were proposed for potential improvements. The creation of a minimum, basic service to be deployed in times of crisis was found to be a good means to improve collaboration between professionals. Regarding training, the use of simulation could be helpful to practice, while classes teaching more theoretical knowledge were also suggested. The creation of working groups or task forces to reflect on the implementation and practical organisation of mental health support was also suggested. In view of the general under-use of the service, a major preoccupation among our respondents was to better identify the needs of HCWs, or the timing of those needs, with a view to providing a more adapted response or making that response available at a more suitable time. Finally, our respondents felt that the barriers to the use of mental health support services among HCWs also warrant investigation.

## 4. Discussion

This study aimed to describe the experiences of healthcare professionals providing mental health support services within their hospital during the first wave of the COVID-19 epidemic in France. To the best of our knowledge, this is the first study to investigate this subject.

There was a strong government-led impetus to provide mental health support services for healthcare professional during the pandemic. Mental health support services targeting HCWs were made available ad hoc during the COVID-19 epidemic, primarily in the form of telephone helplines due to distancing restrictions. These services were staffed by volunteers with expertise in mental health who were temporarily unable to continue their own professional activity due to the redeployment of hospital resources to deal with the COVID-19 epidemic. Our study showed that the volunteers staffing mental health support services were mainly women, aged 35 to 39 years, and predominantly psychologists with more than 10 years’ professional experience. This female predominance is unsurprising and is in line with the gender distribution of the psychological sciences [10]. Despite this long experience working in mental health, the vast majority of respondents (86.5%) had never previously worked on a helpline of this sort, although they felt comfortable in this role as shown by the median score of eight on the self-reported comfort level evaluation. They were also largely in favour of doing it again if needed. Taken together, these findings underline that mental health professionals were willing to take part in the government-led effort to provide mental health support for healthcare workers. They participated enthusiastically and were quite satisfied overall with the experience, despite having encountered some difficulties.

In our study, the respondents reported difficulties with various aspects of the job, specifically the practical organisation, the existence of clinical situations that were unrelated to COVID-19, and the frustration of having callers who refused to accept the help on offer. The respondents remained positive nevertheless, proposing several suggestions for future improvements, should the service be maintained or repeated. Our proposals for future improvements are summarised in Table 3.

Regarding the organisation, the intention of the health authorities in requesting the urgent implementation of this service was to model it on the existing medico-psychological emergency units [4], which exist in France to respond to large-scale emergencies such as terrorist attacks or natural disasters. The mental health support for the epidemic was intended to be temporary, and the services were staffed by volunteers with qualifications and expertise in mental health. Although modelled on the medico-psychological emergency units, only one-third of the COVID support services actually worked in collaboration with them, which is surprising. Indeed, medico-psychological emergency units are the gold standard for large-scale events with strong psychological repercussions. One might therefore think that the COVID-19 epidemic could be considered just such an event in view of its extent and its impact on the mental health of HCWs, particularly the almost daily confrontation with death [5]. Existing medico-psychological emergency units could thus have been considered as the reference in terms of mental health support.

Using existing structures such as the medico-psychological emergency units would also have avoided the problems encountered by groups of disparate professionals trying to work together without clear hierarchical links or work structures. Indeed, medico-psychological emergency units have a common structure, with established procedures for collaboration between professionals and between units to enable the deployment of large-scale services in the case of mass emergencies [4].

In spite of this, in our study, the vast majority of the service providers felt comfortable or very comfortable providing this service, notably due to solid clinical experience, qualifications, and competence in the field, with many having training in emergency care. This is congruent with literature reports showing that being competent enhances the comfort of performing the task [11,12]. This provides an additional argument in favour of involving medico-psychological emergency teams in the management of hospital staff during times of major healthcare crisis.

The persistence of the epidemic and the progressive return to usual activities since May 2020 have raised the question of how long the mental health support services should continue to be offered. Indeed, the volunteers staffing the service have now returned to their usual work, and the mental health support services are for the most part no longer available. Yet, the epidemic continues, and HCWs in the hospital setting continue to suffer from a high level of tension due to the continuing influx of COVID patients while at the same time trying to catch up for lost time on other activities that were cancelled for long periods at the start of the epidemic. This all comes on top of the usual volume of work, putting HCWs under considerable strain. Unfortunately, many mental health support services that were created ad hoc for the pandemic were discontinued as soon as the peak passed, whereas many healthcare professionals continue to experience a need for support.

The uptake of the mental health support services was generally quite low. Regarding the clinical situations encountered by the respondents in our study, they were sometimes surprised to find that callers reached out to them for problems unrelated to the COVID-19 epidemic. A number of callers to the service had pre-existing psychological, psychiatric, or social problems that were exacerbated by the crisis. This included problems in the workplace, and exacerbation by the lockdown of mood disorders [13,14,15], alcohol consumption [16], or domestic violence [17,18,19]. Furthermore, respondents in our study were frustrated and unsettled by callers who refused help as well as by the failure of HCWs to make use of the mental health support service made available. Nevertheless, this under-use is in line with literature data [20,21,22] reporting that HCWs had difficulties reaching out for help, despite the documented existence of mental health disorders [23,24,25,26]. For some of the service providers, the under-use of the service was frustrating. It compounded their feeling that the service being provided was not sufficiently well recognised by their peers (who failed to use it) and by the hospital administration (who failed to advertise it enough). Conversely, other respondents underlined the importance of investigating the reasons behind the low rate of use.

We believe there are several possible explanations for the low rate of uptake. It is primarily the result of a temporal mismatch, whereby the help was provided for the duration of the first wave when HCWs were too busy and exhausted working. As soon as the first lockdown was lifted and normal activities were resumed in May 2020, the mental health support services were discontinued, at a time when many HCWs might have been most in need of them. It is therefore possible that the HCWs did not have the practical or psychological availability to contact the service while it was on offer during the peak of the epidemic. It should be noted that hospitals in the eastern part of France were hardest hit, with hospitals full to capacity. The national emergency preparedness plan was in action, which allows for the mobilisation of staff 24/7. Many of the mental health support services were only available during the daytime, thereby limiting the possibilities for HCWs to contact them. Secondly, this form of help may not have been considered useful by the HCWs who were the target audience. The individualised mental health support was provided over the telephone, with volunteers who were not familiar with the intensive care or acute care environments. The HCWs may not have found the help offered to be appropriate to their needs. A more holistic approach to well-being in the workplace might be more suitable than targeting individual workers. This is key to understanding professional practices in routine daily practice and the related emotional and relational difficulties [27,28]. To this end, support from the hierarchy and recognition from the hospital administration are essential [29]. The opportunity to debrief frontline HCWs is also an interesting avenue to investigate [30]. It has been reported that help is more readily accepted when it is offered by persons who are known to the healthcare team or who work in the same environment [31]. In this regard, partnership with the existing medico-psychological emergency units might have helped to facilitate contacts with the new mental health support services. Placing them under the same umbrella as an existing service would have helped to make them more widely known.

Mental health support services may not have been the solution most adapted to the needs of HCWs during the pandemic. A Cochrane review published in 2020 investigated the workplace interventions implemented to support the mental health and resilience of frontline health and social care professionals during the COVID crisis [32]. They underlined that simple, flexible, and adapted solutions are best, and it is also necessary to raise awareness among HCWs of the need to be attentive to their own health needs, particularly mental health. Finally, training in management of crisis situations is also important.Other authors [28,33] have purported that well-being in the workplace during this type of crisis is more likely to be achieved by means of material assistance (such as “down-time” in specifically equipped rest areas or logistic aid to make daily life more comfortable). This could be another potential explanation for the under-use of the mental health support services, since the HCWs may have been more in need of material rather than psychological support.

In low-resource settings, attempts to use tele-mental health services to reach underserved groups were found to be feasible but were not necessarily suitable for all types of patients and situations [34]. A large-scale meta-analytic atlas including 173 studies (n = 502,261 individuals) and investigating mental health problems during the COVID pandemic reported that anxiety, depression, and post-traumatic symptoms were more prevalent in LMICs, although this was not specifically a study of HCWs [35]. A recent systematic review and meta-analysis of the prevalence of mental health conditions specifically in HCWs during and after a pandemic highlighted the paucity of data from LMICs [36]. Nevertheless, many LMICs, particularly those in Africa, have previous experience of epidemics, such as Ebola virus, or major traumatic events (e.g., Rwandan genocide of 1994), which have promoted resilience and a culture of mental health support. Indeed, in Rwanda, for example, policies implemented in the wake of historical traumatic events meant that an organised system was already in place and could be rapidly deployed to face the COVID crisis [37].

Our study has some limitations. Firstly, we identified 36 hospitals with mental health support services, and from these, a total of 37 providers responded to the questionnaire. This may appear to be a low response rate. However, the exact numbers of staff in each support service were unknown to us, thus precluding the calculation of a more exact rate. Secondly, although the respondents in our study presented characteristics similar to the general population of HCWs, notably in terms of gender distribution, they may not be representative of the overall population of providers of mental health support services in our country. We cannot rule out the possibility that only providers who have a particular interest in research or in their work responded, leaving room for potential selection bias. Furthermore, since our point of contact with each service was through the chief, participants may have perceived hierarchical pressure to respond.

Conversely, our study also has some strengths. It is an innovative study that provides unique insights into the functioning of mental health support services in a crisis. It opens avenues for future reflection on the possible improvements that could be made to better support hospital staff. The impact of the current crisis on the mental health of HCWs has been well documented [5,7] and is alarming. In this context, questioning hospital staff about their experience and perception of the crisis and their personal and professional resources and difficulties is an indispensable step towards identifying specific aspects that could be leveraged to improve existing services or targeted for future preventive measures. Finally, other avenues for future research include investigations of the long-term persistence of mental health disorders in HCWs or studies of opportunities for providing mental health support services in the long term.

## 5. Conclusions

In France, many hospitals set up mental health support services to deal with the mental health repercussions of the COVID-19 epidemic for HCWs. These services were staffed by volunteers, mainly women, aged 35 to 49 years with over 10 years’ experience. However, the rate of use of these mental health support services was low. This suggests that further modifications are warranted to ensure that the response to periods of exceptional pressure on healthcare services is adapted to the needs of the hospital staff, both in terms of the content of the service and the timing of its delivery. Further research is necessary to identify the needs of HCWs and potential barriers to the use of mental health support services during times of healthcare crises.

## Figures and Tables

**Figure 1 healthcare-10-01337-f001:**
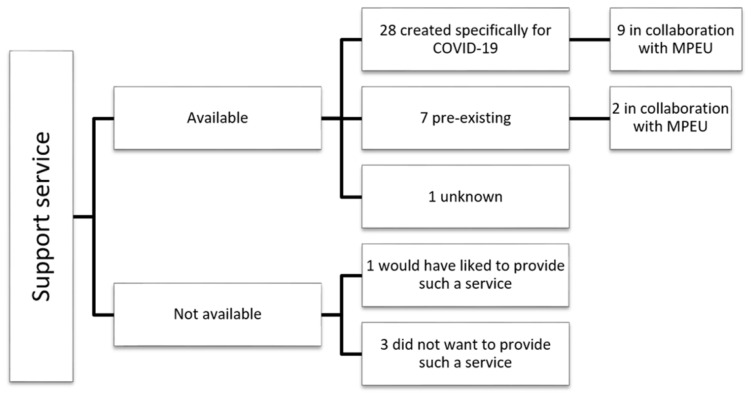
Details of the mental health support services in place during the first wave of the COVID-19 epidemic in France, according to study respondents. MPEU, medico-psychological emergency unit.

**Table 1 healthcare-10-01337-t001:** Characteristics of the 36 mental health support services identified among 77 centres participating in the PsyCOVID-ICU study.

Staffed by…	N	%
**Psychologists**	**35**	**95.7**
From within the same hospital only	28	80
From outside establishments	2	5.7
From private practice only	-	-
Mixed	5	14.3
**Psychiatrists**	**17**	**47.2**
From within the same hospital only	14	82.3
From outside establishments	1	5.9
From private practice only	1	5.9
Mixed	1	5.9
**Target audience of the support services**		
Healthcare workers only	7	19.4
Healthcare workers and patients’ families	27	75
Unknown	2	5.6
**Healthcare workers targeted**		
Intensive care healthcare workers only	2	5.6
All healthcare workers in the hospital	23	63.8
All healthcare workers inside and outside the hospital (e.g., private practice, nursing homes)	9	25
Unknown	2	5.6
**Means of contact**		
Telephone call centre	33	91.7
Web-based service	3	8.3

**Table 2 healthcare-10-01337-t002:** Characteristics of the healthcare workers staffing the mental health support services (N = 37).

Region of Work	N	%
Greater Eastern region of France	21	57
Rest of Metropolitan France	16	43
**Sex**		
Female	28	76
Male	9	24
**Age category**		
20–34 years	9	24.3
35–49 years	16	43.3
50–65 years	12	32.4
**Profession**		
Psychologist	25	67.6
Physician	8	21.6
Nurse	4	10.8
**Number of years of professional experience**		
<5 years	6	16.2
5 to 10 years	5	13.5
>10 years	26	70.3
**Usual place of work**		
Hospital	36	9.3
Specialised psychiatric hospital	8	22.2
University teaching hospital	19	52.8
Not specified	9	25
Missing data	1	2.7
**Specific training in…**		
Medico-psychological emergencies	18	48.6
Occupational health	4	10.8
**First time working on a helpline**		
Yes	32	86.5
No	5	13.5
**I would do it again**		
Yes	35	94.6
No answer	2	5.4
**Additional training would be useful to me**		
Yes	22	59.5
No	10	27.0
No answer	5	13.5

**Table 3 healthcare-10-01337-t003:** Proposals for future improvements to mental health support service provision.

Services offered
Set up a service that can be activated quickly when required, with appropriate publicity to make the service known to the target audience Comprising reference persons and staff who can be mobilised quickly in case of a major crisis Also able to mobilise personnel over the long term
**Daily functioning**
In collaboration with medico-psychological emergency units In collaboration with the hospital’s occupational health department In collaboration with networks of psychiatric health professionals in the region to enable referral and follow-up of healthcare workers with acute needs
**Access/Availability**
By telephone or presence-based, depending on the type of healthcare crisis Flexible hours to enable access for all healthcare workers (day and night staff)
**Care provided**
Help should be on offer throughout the full duration of the healthcare crisis Alternative forms of management should be offered: team debriefing, informal exchange in groups, therapeutic activities Material assistance (provide equipment, etc.)
**For the persons staffing the service**
Provide opportunities for interprofessional communication Provide opportunities for support and debriefing Provide specific training, possibly in the form of roleplay or simulation

## Data Availability

The datasets used and/or analysed during the current study are available from the corresponding author on reasonable request.

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
