# Peer review of "Mental Health Support for Hospital Staff during the COVID-19 Pandemic: Characteristics of the Services and Feedback from the Providers"

_healthcare, 2022, doi:10.3390/healthcare10071337_

Round 1
Reviewer 1 Report
The sample size of the study can be enlarged to draw a more reliable outcome.
Author Response
Thank you for Reviewing our work. We agree that a larger sample might yield additional information but unfortunately, for the purposes of the present study, the work is now completed. It is not possible to increase the sample size now after the fact. We will plan for larger samples in our future works on similar topics.
Reviewer 2 Report
The current manuscript deals with mental health support during COVID-19 pandemic and this is point of high interest, but there are major points that need to be addressed:
1- Authors must put the ethical approval number and committee that approve this work , It is a must.
2- The part of methodology (Study design) , authors must put graphical chart that clarify the methods used and study design to clear stages for readers.
3-Authors must focus in writing their discussion and add key points to clarify their work.
4-Authors must state the clear point of significance of work , it is needed to give total message to readers.
Author Response
The current manuscript deals with mental health support during COVID-19 pandemic and this is point of high interest, but there are major points that need to be addressed:
Thank you for your thorough review and useful comments.
1- Authors must put the ethical approval number and committee that approve this work , It is a must.
Thank you for pointing out this omission. The details of the Ethics Committee approval have been added to the methods section (Section 2.1.1).
2- The part of methodology (Study design) , authors must put graphical chart that clarify the methods used and study design to clear stages for readers.
Thank you for this useful suggestion. A graphical presentation of the methods has been added to the Supplementary Material.
3-Authors must focus in writing their discussion and add key points to clarify their work.
4-Authors must state the clear point of significance of work , it is needed to give total message to readers.
Thank you for these suggestions. We have introduced summary sentences to underline the key points of the main paragraphs in the discussion.
Reviewer 3 Report
This article addresses how COVID related situations mentally affected health care support providers in France. The reviewer believes that this study can contribute to making policies or supports for health-related employees in France. Even though the method is not scientifically grounded, the current study provides with empirical knowledge under a specific circumstance in France. The following is my recommendation to improve the article.
1. The order or title in "2. Method" section is quite confusing. For instance, 2-3, the authors entitled the section as "Stage 1, 2, 3..." - In general, when the title is labeled as "stages," it means there would be gradual changes or a development in processes as the number goes high. However, it seems the contents of each stage are independent. Thus, I would recommend the authors remove "stage" from the title.
2. The conclusion indicated that there should be support for healthcare provider considering stress and challenges under COVID outbreak (and also addressed some examples). I am wondering whether the conclusion can be generalized in the future after the outbreak is over since it is too specific circumstance.
I appreciate the authors who gave me the chance to review this valuable article.
Author Response
Reviewer 3
This article addresses how COVID related situations mentally affected health care support providers in France. The reviewer believes that this study can contribute to making policies or supports for health-related employees in France. Even though the method is not scientifically grounded, the current study provides with empirical knowledge under a specific circumstance in France. The following is my recommendation to improve the article.
Thank you for your suggestions for improvement.
- The order or title in "2. Method" section is quite confusing. For instance, 2-3, the authors entitled the section as "Stage 1, 2, 3..." - In general, when the title is labeled as "stages," it means there would be gradual changes or a development in processes as the number goes high. However, it seems the contents of each stage are independent. Thus, I would recommend the authors remove "stage" from the title.
Thank you for pointing this out. The term “stages” has been removed from the methods and the results sections.
- The conclusion indicated that there should be support for healthcare provider considering stress and challenges under COVID outbreak (and also addressed some examples). I am wondering whether the conclusion can be generalized in the future after the outbreak is over since it is too specific circumstance.
We have reformulated to indicate that these findings and possible avenues for improvement apply to any period of exceptional pressure on the healthcare services.
I appreciate the authors who gave me the chance to review this valuable article.
Thank you for your valuable remarks.
Reviewer 4 Report
Dear Authors,
Difficulties encountered in mental health provision during the epidemic have not been quantified, so the significance of responses cannot be assessed. The data analysis section states that this assessment will be performed in relation to the themes identified,but in the end this has not been reflected in the results.
The findings partially answer the research question aimed.
1. The main question addressed by the research is to identify key areas for improvement in relation to the provision of mental health support during COVID-19. Regarding the qualitative part, the degree of response to the difficulties encountered by the respondents is not specified. It is not possible to determine whether that particular aspect of the 49 situations that were identified was indicated by one respondent, which would be insignificant, or by a relevant number, which would imply a greater need for attention.
2. Given the exceptional pandemic situation, it is a current issue, but as the discussion demonstrates, addressed for other crisis situations. It is considered that the introduction, on the basis of the referenced bibliography, does not provide a broader view.
3. The main contribution is the identification of difficulties by established support services in addressing the psychological support of health professionals in a pandemic situation.
4. In addition to classifying by themes, the weighting of each difficulty should be specified for better understanding.
5. The conclusions are consistent with the evidence and arguments presented but do not address the main issue posed.
6. The references are appropriate.
7. The tables and figures represent well the characteristics of the support services and their members. It would be useful to include some graphical support in relation to the qualitative analysis.
Author Response
Reviewer 4
Difficulties encountered in mental health provision during the epidemic have not been quantified, so the significance of responses cannot be assessed. The data analysis section states that this assessment will be performed in relation to the themes identified,but in the end this has not been reflected in the results.
Thank you for your review and suggestions. The main findings of this study derive from a qualitative study, so by definition, there is no quantification, but rather a qualitative analysis of the discourse of the participants. The quantitative description of the centres identified, the number of people contacted, and the final number of participants are detailed at the beginning of the methods, and in the flowchart in Figure 1.
Regarding the themes: the explanation of major and minor themes given in the methods is to inform the reader about how thematic analysis is performed. In the results, the themes that arose from our study participants are detailed in section 3.3 (Feedback from the services providers), and as the headings indicate, these themes were: (1) working on mental health support services; and (2) difficulties encountered during the provision of these services, which in turn is subdivided into the different types of difficulties. The interview guide in qualitative analysis is just that: a guide. The objective of qualitative enquiry is to allow the participants to speak freely about the topics that are most important to them. So it is not unusual for the themes arising from the analysis to differ from those of the interview guide.
We have done our best to describe the findings in a logical sequence, within the limits of the word count allowed by the journal.
- The main question addressed by the research is to identify key areas for improvement in relation to the provision of mental health support during COVID-19. Regarding the qualitative part, the degree of response to the difficulties encountered by the respondents is not specified. It is not possible to determine whether that particular aspect of the 49 situations that were identified was indicated by one respondent, which would be insignificant, or by a relevant number, which would imply a greater need for attention.
This is indeed an important point. We have added the details of the number of times each theme was mentioned in a new Additional File (Additional File 3, cited in section 3.3.2.).
- Given the exceptional pandemic situation, it is a current issue, but as the discussion demonstrates, addressed for other crisis situations. It is considered that the introduction, on the basis of the referenced bibliography, does not provide a broader view.
Thank you for this perspective. We have rephrased the conclusion to indicate that the suggestions for improvement and experiences are relevant for any periods of exceptionally high pressure on the healthcare services.
We have also introduced a short reference to previous outbreaks of contagious disease in the introduction to indicate that the anxiety arising from such situations has previously been demonstration. However, the unprecedented scale of the current COVID pandemic means that even existing studies from previous outbreaks cannot come close to addressing the full spectrum of issues encountered in the COVID era.
- The main contribution is the identification of difficulties by established support services in addressing the psychological support of health professionals in a pandemic situation.
Thank you for this accurate summary. We believe that these points could be helpful in improving the provision of mental health services in the future, not only in the context of a healthcare crisis, but even during “ordinary” times.
- In addition to classifying by themes, the weighting of each difficulty should be specified for better understanding.
Thank you for this remark – as indicated in the response to Comment number 1 above, we have added the details of the number of times each theme was mentioned in a new Additional File (Additional File 3, cited in section 3.3.2.).
- The conclusions are consistent with the evidence and arguments presented but do not address the main issue posed.
Thank you for your assessment. We agree that the question of how to improve these services remains open, and may depend on how receptive different hospital administrations are to changes in their methods of delivering mental health services. Other factors such as government impetus, funding and availability of qualified staff may also influence the ability of these services to be modulated in accordance with our findings.
- The references are appropriate.
Thank you.
- The tables and figures represent well the characteristics of the support services and their members. It would be useful to include some graphical support in relation to the qualitative analysis.
Thank you for your appreciation. As outlined above, a new Additional Table has been added with the distribution of the different themes across respondents.
Round 2
Reviewer 1 Report
The paper is well structure.
Reviewer 4 Report
Dear Authors,
The changes made to the original document are considered to have increased the significance of the paper's contributions.